# Neural flip-flops I: Short-term memory

Lane Yoder *

Department of Science and Mathematics, University of Hawaii, Honolulu, Hawaii, United States of America

* LYoder@hawaii.edu

## Abstract

The networks proposed here show how neurons can be connected to form flip-flops, the basic building blocks in sequential logic systems. The novel neural flip-flops (NFFs) are explicit, dynamic, and can generate known phenomena of short-term memory. For each network design, all neurons, connections, and types of synapses are shown explicitly. The neurons' operation depends only on explicitly stated, minimal properties of excitement and inhibition. This operation is dynamic in the sense that the level of neuron activity is the only cellular change, making the NFFs' operation consistent with the speed of most brain functions. Memory tests have shown that certain neurons fire continuously at a high frequency while information is held in short-term memory. These neurons exhibit seven characteristics associated with memory formation, retention, retrieval, termination, and errors. One of the neurons in each of the NFFs produces all of the characteristics. This neuron and a second neighboring neuron together predict eight unknown phenomena. These predictions can be tested by the same methods that led to the discovery of the first seven phenomena. NFFs, together with a decoder from a previous paper, suggest a resolution to the longstanding controversy of whether short-term memory depends on neurons firing persistently or in brief, coordinated bursts. Two novel NFFs are composed of two and four neurons. Their designs follow directly from a standard electronic flip-flop design by moving each negation symbol from one end of the connection to the other. This does not affect the logic of the network, but it changes the logic of each component to a logic function that can be implemented by a single neuron. This transformation is reversible and is apparently new to engineering as well as neuroscience.

## 1. Introduction

This article is the fourth in a series of articles that show how neurons can be connected to process information. The first three articles [1–3] explored the analog properties of neuron signals in combinational logic operations, whose outputs depend only on the current state of the inputs. A fuzzy logic decoder was shown to generate the major phenomena of both olfaction and color vision (such as color mixing, mutually exclusive colors, and the shape of perceived color space), including the brain's shortcomings (such as the Bezold-Brücke hue shift) [1, 2]. The decoder's design is radically different from a standard electronic digital (Boolean logic) decoder [2, 4, 5]. If implemented with electronic components and given digital inputs, the decoder performs the same Boolean function as the standard digital design more efficiently.

**Data Availability Statement:** The novel networks proposed in the manuscript are theoretical. There are no data.

**Competing interests:** The authors have declared that no competing interests exist.

It was shown that a single neuron with one excitatory input and one inhibitory input, with signal strengths X and Y, respectively, can function as a logic primitive, X AND NOT Y [1, 2]. In simplest terms, this is because the neuron is active when it has excitatory input *and* does *not* have inhibitory input. It was also shown that an AND-NOT gate can be configured to function as an inverter (i.e., a NOT X logic primitive). The AND-NOT gate together with a NOT gate make up a functionally complete set, meaning any logic function can be performed by a network of such components. The neuron AND-NOT gate will be reviewed here and used in the proposed networks.

The present article considers the Boolean logic properties of neuron signals in sequential logic operations, whose outputs are functions of both the current inputs and the past sequence of inputs. That a neuron can operate as a functionally complete logic gate, analog or digital, provides a framework for the brain's processing of information—analog and digital, combinational and sequential.

Flip-flops are the basic building blocks of sequential logic systems. A flip-flop is a mechanism that can be set repeatedly to either one of two stable states, commonly labeled 0 and 1. A flip-flop can be used as a memory mechanism to store one bit of information. It is shown here that a few AND-NOT gates can be connected to perform the same function as two standard electronic flip-flops, an active low and an active high Set-Reset (SR) flip-flop. These are not the only flip-flops that can be constructed with AND-NOT gates, but they may be the simplest. The network designs are modifications of standard electronic logic circuit designs. It is shown here that the NFF designs are derived directly from the standard electronic designs simply by moving each negation circle from one end of the connection to the other. This changes the logic of each component, but it does not materially affect the logic of the network. The modifications are necessary to implement the circuits with neurons because the AND-NOT gate is virtually never used as a building block in electronic computational systems.

The NFFs produce both known and testable, unknown phenomena of short-term memory. With inputs from the outputs of NFFs, neural decoders proposed in [2] can retrieve encoded information that is held in NFFs. That is, a memory can be recalled. The NFFs' robust operation in the presence of noise is demonstrated here by simulation, but the properties can be proven directly from the explicit network connections and minimal neuron properties of excitation and inhibition. In [6] it was shown that NFFs, together with a network that can produce the oscillations commonly known as brainwaves, suggest a resolution to the longstanding controversy of whether short-term memory depends on neurons firing persistently or in brief, coordinated bursts [7, 8].

The NFFs' operation is dynamic, meaning the only changes are the levels of neuron activity. No structural change is required, such as neurogenesis, synaptogenesis, or pruning, nor is any change required in the way neurons function, such as a change in synaptic strength or the strength of action potentials. This makes the networks' speed consistent with the "real time" of most brain functions (a few milliseconds). The NFFs' architectures are explicit, meaning all neurons, connections, and types of synapses are shown explicitly, and all assumptions of neuron capabilities are stated explicitly. Only minimal neuron capabilities are assumed, and no network capabilities are assumed.

It was shown in [9] that designing a simple logic circuit that can perform a single, biologically advantageous task can lead to a discovery of how neurons are connected to process information. This is the method that was used to find the networks proposed here and in [1–6]. Besides performing a biologically useful task, the networks are dynamic, explicit, and able to generate phenomena that are central to a particular brain function. These four properties are characteristics that networks in the brain must have. The neuron properties used to achieve

the results for these networks—excitation, inhibition, and sigmoid neuron responses—have been known a long time.

## 2. Unexplained phenomena and previous models

### 2.1. Single neuron logic capability

McCulloch and Pitts' seminal paper [10] proposed that the brain is made up of logic gates. The idea of Boolean neurons had a tremendous effect on artificial neural networks and machine learning, but it had a limited impact on neuroscience [11]. More than 70 years later, the brain's computational capabilities are still unclear [12]. In that time span, many theoretical models have been proposed for neuron responses as mathematical or logic functions, but the modern view follows "the adage that all models are wrong, but some are useful" [13].

The neuron response model proposed in [1, 2] demonstrated that a neuron with one inhibitory input that can suppress one excitatory input can function as an AND-NOT gate, and that this logic primitive is sufficient for all logic operations. This demonstration was apparently the first claim that a single neuron can function as a specific logic primitive based on minimal neuron capabilities of excitation and inhibition. This neuron response model will be reviewed and used here.

### 2.2. Short-term memory

Memory tests have shown that certain neurons fire continuously while information is held in short-term memory. This activity was found in neurons in the visual, auditory, and sensorimotor cortexes of monkeys while corresponding sensory information is held in memory [14, 15]. Similar activity has been found more recently in humans [16].

In the first experiments [14, 15], seven characteristics of neural activity were associated with memory formation, retention, retrieval, termination, and errors: 1) Before the stimulus was presented, the sampled neuron discharged at a low, baseline level. 2) When the stimulus was presented, or shortly after, the neuron began to fire at a high frequency. 3) The high frequency firing continued after the stimulus was removed. 4) The response was still high when the memory was demonstrated to be correct. 5) The response returned to the background level shortly after the test. 6) In the trials where the subject failed the memory test, the high level firing had stopped or 7) had never begun.

It will be shown that the memory bank of NFFs presented here produces all of these phenomena.

### 2.3. Previous models

Considerable progress has been achieved for long-term memory, notably with models based on synaptic strength changes [e.g., 17–20]. Models of single neuron logic gates [e.g., 21] and short-term memory mechanisms composed of neurons [e.g., 21–26] have multiple problems.

Much of the literature on possible mechanisms for memory in the brain concentrates on observed changes in the nervous systems of various organisms and says little about the characteristics necessary for these changes to serve as memory. Since changes occur continually throughout the body for many reasons, change by itself is weak evidence of memory formation.

If a change is to serve as memory, it must be capable of representing information and there must be a means of retrieving that information. If the mechanism is to be more flexible than a permanent storage space, there must be a way to replace the information stored there. In addition, a robust and practical memory device should be capable of storing different kinds of

information; it should be inexpensive in resource requirements; storing, retrieving, and changing information should be simple, reliable, and fast; information should remain stored reliably and unambiguously until it is no longer needed or until new information replaces it; and errors should be minimal and correctable. The absence of one or more of these properties is the reason such changes as muscle growth after exercise are not plausible memory devices.

These characteristics are not obvious in most memory models that have been proposed. On the contrary, most models appear to be incapable of many of them. Yet even the most basic requirements for a memory mechanism are routinely ignored in the literature.

Most models of neuron response functions and short-term memory mechanisms composed of neurons are speculative, needlessly complex, and do not include evidence or even a plausible argument that the proposed mechanisms would operate as claimed. Some network models are simply "black boxes" with no evidence that they can actually be implemented with neurons. Neurons and connections are seldom shown explicitly. Some models make tacit assumptions of powerful neuron capabilities. When assumptions of neuron capabilities are stated, supporting evidence is not included.

Except for the models presented here, networks that are dynamic, explicit, and can produce known phenomena of short-term memory are virtually nonexistent. At a minimum, these properties are necessary for a realistic model of short-term memory. As two of the most plausible examples of other models, claims in [21] of a single neuron logic gate and an explicit network that can function as a flip-flop are discussed here in some detail. Reviewing more models that are not dynamic and not explicit and do not produce known phenomena would not serve any useful purpose.

**2.3.1. Threshold oscillator neuron response model.**   The neuron response model in [21] is a "threshold oscillator." This means that for an excitatory input strength below a certain threshold, the neuron has little or no response, and for inputs at or above the threshold, the neuron spikes at a high rate.

**2.3.2. AND gate.**   The authors of [21] claim that with the threshold oscillator model for neuron signals, a single neuron can function as a logic AND gate. The AND gate neuron has two (or presumably more) excitatory inputs representing logic values TRUE or FALSE.

The claim for the AND gate neuron is based on two assumptions of finely tuned input strengths. If all of the inputs represent the logic value TRUE, they are 1) assumed to be sufficiently high for the combined input to reach or surpass the neuron's threshold, producing a high output representing the logic AND value TRUE. The TRUE inputs are also 2) assumed to be sufficiently low so that if one of the inputs represents the truth value FALSE, the combined input is below the gate's threshold, producing a low output representing the AND value FALSE.

This logic gate model has at least two problems. Although the paper's neuron response model is said to be the threshold model, the AND gate's input neurons do not produce the high signals of the threshold model. The high input values representing the logic value TRUE are each necessarily below the AND gate neuron's threshold. Second, no evidence or argument is given for how the input neurons can maintain the signals of intermediate strengths representing TRUE with enough precision (high enough to surpass the threshold together, but not high enough to surpass the threshold if one is low) to produce the claimed outputs.

In contrast, the design of the AND gate implemented with AND-NOT gates follows from straightforward logic because the AND-NOT gate is functionally complete:

$$X \text{ AND } Y = X \text{ AND NOT (NOT } Y).$$

The AND gate can be implemented with two AND-NOT gates: a first AND-NOT gate configured as an inverter (Fig 4B) that provides input to a second AND-NOT gate.

**2.3.3. Fitzhugh Nagumo set reset flip-flop.** This simple flip-flop model [21, Fig 10] consists of two neurons with reciprocal inhibitory input and continuously high excitatory input to each cell. Each cell has an additional excitatory input (Set and Reset) that is variable and normally low.

The model's operation is based on four assumptions. The flip-flop is 1) assumed to be initially in a stable state, with the inhibitory input from one cell 2) assumed to inhibit the continuous high input to the other cell, leaving the first cell with no inhibitory input and a high output representing one bit of stored information. The flip-flop state is inverted with a brief, high excitatory input to the second, inhibited cell. The combined two high inputs are 3) assumed to be sufficiently high to override the inhibitory input and produce an inhibitory signal to the first cell. This inhibition is 4) assumed to be sufficiently high to suppress the first cell's excitatory input, thus switching the outputs. When the brief high excitatory input to the second cell ends, the flip-flop is in a stable state with the second cell inhibiting the first.

This flip-flop model has several problems. How the flip-flop is initialized in a stable state without producing a race condition is not discussed. The model's operation is demonstrated by simulation with electronic components, but no evidence or argument is given that indicates neurons are capable of the four assumptions of somewhat complex behavior.

## 2.4. Testable predictions of unknown phenomena

This article ends with several testable predictions that are implied by the models, briefly outlined here. Since the proposed networks are explicit, any of them can be constructed with actual neurons and tested for specific predicted behaviors.

As noted above, one of an NFF's two outputs produces all seven characteristics of neuron activity while information is held in short-term memory. NFFs predict eight additional phenomena for the pair of outputs. These predictions can be tested by the same methods that led to the discovery of the first seven phenomena. The two NFF output neurons are predicted to have 1) close proximity; 2) reciprocal, 3) inhibitory inputs; 4) complementary outputs; and 5) noise-reducing responses to the inputs. When the memory state is changed, 6) the neuron with high output changes first with 7) the other changing a few milliseconds later. 8) After the memory test, the outputs of both neurons are low.

## 3. Simulation methods

A neuron's response to an excitatory input of strength X and an inhibitory input of strength Y is represented by the function F(X, Y). The response function's minimal noise reducing properties that can produce the network properties claimed here are given in inequalities 1 and 2, section 4.2.2.1 below. These conditions generalize the noise-reducing properties of a sigmoid function. (A sigmoid response reduces moderate levels of additive noise in a binary input signal by producing an output that decreases a low input and increases a high input.) An example of a neuron response function that satisfies these conditions is given in section 4.2.2.2. The graphs of this function and the associated plane in Fig 2B were created in MS Excel and MS Paint. The graph in Fig 2A was created with Converge 10.0.

This example neuron response function F(X, Y) was used to simulate the NFF shown in Fig 4F. The simulation was done in MS Excel as follows. The number $t_i$ represents the time after i neuron delay times, i = 0, 1, 2,. . .. At time $t_0$, the NFF's neurons are initialized in a stable state. Simulated inputs to the NFF are given. At time $t_i$ for i > 0, the output $Z_i$ of each NFF neuron

that has excitatory and inhibitory inputs $X_{i-1}$ and $Y_{i-1}$ at time $t_{i-1}$ is $Z_i = F(X_{i-1}, Y_{i-1})$. The graphs of the simulated NFF inputs and outputs are shown in Fig 5.

A simulation using a specific neuron response model can support network claims only for that model. This study goes substantially further. As stated above, the inequalities 1 and 2 in section 4.2.2.1 are the minimum neuron requirements to produce the NFF results in the presence of noise. All of the claims for the NFF's sustained binary outputs in the presence of noise can be proven (somewhat tediously) from the two properties and the network architecture in Fig 4F. Therefore the network results are verified for any neuron response function that satisfies these two inequalities.

A single-transistor AND-NOT gate is shown in section 4.2.2.3, Fig 3A, to demonstrate that the two noise-reducing properties do not indicate capabilities of sophisticated mathematics. The figure was created and simulated in CircuitLab. The graphs of its response function and related plane in Fig 3B were created in MS Excel and MS Paint.

## 4. Analysis

### 4.1. Figure symbols

For several reasons, the neural networks in the figures are illustrated with standard (ANSI/IEEE) logic symbols rather than symbols commonly used in neuroscience schematic diagrams. A comparison is shown in Fig 1.

The symbols in Fig 1A can be interpreted in two ways. As a logic symbol, the rectangle with one rounded side represents the AND logic function, and a circle represents negation. So the networks in the figures can be constructed with ordinary electronic components or simulated with electronic circuit software. Second, it will be shown that the logic gate represented by an AND symbol and a circle can be implemented by a single neuron, with a circle representing inhibitory input and no circle representing excitatory input. As shown in Fig 1B, neurons are often represented by a circle, inhibition by a small closed circle, and excitation by a closed triangle, but there does not seem to be an accepted standard of symbols for networks of neurons.

The standard logic symbols normally represent Boolean logic, which for most electronic computational systems means digital signal processing. Neurons can convey analog signals, either with signals of graded strength or with the strength of signals consisting of spikes measured by spike frequency. It will be shown that the neural networks in the figures can generate robust digital signals, i.e., signals with only high and low strengths (except during transition from one state to the other).

The similarities and differences between the novel diagrams of networks that can be implemented with neurons, and diagrams of standard logic circuits for the same functions implemented electronically, are easier to see if they are both illustrated with the same symbols.

The single, branching output channels in Fig 1A are more realistic depictions of most axons than the multiple output channels of Fig 1B.

Finally, diagrams in standard engineering form clarify the connectivity, the type of each connection, the logic function of each component, the distinction between feedback (right to left) and feed-forward (left to right) signals, and the overall direction of a network's signal processing from input to output (left to right).

### 4.2. Neuron signals

All results for the networks presented here follow from the neuron response to binary (high and low) input signals, given in the next section, and the algebra of Boolean logic applied to the networks' connections. Although binary signals are common in modeling neuron response, how neural networks are capable of maintaining binary outputs in the presence of

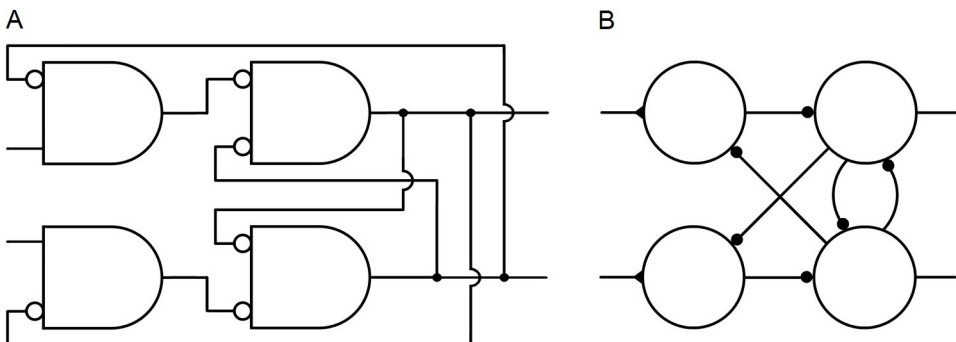

**Fig 1. Network symbols. A.** A logic circuit illustrated with standard logic symbols. Each of the four components represents a logic function that can be implemented with electronic hardware or with a single neuron. **B.** The same logic circuit illustrated with symbols commonly used in neuroscience schematic diagrams.

additive noise in binary inputs has apparently not been demonstrated. Analog signals (intermediate strengths between high and low) are considered here only to show how the networks in the figures can generate robust binary signals in the presence of moderate levels of additive noise.

**4.2.1. Binary neuron signals.** Neuron signals can be graded or they can consist of all-or-nothing action potentials, or spikes. As noted above, the strength, or intensity, of a signal consisting of spikes can be measured by the spike frequency. Neuron signal strength is normalized here by dividing it by the maximum strength for the given level of adaptation. This puts intensities in the interval from 0 to 1, with 0 meaning no signal and 1 meaning the maximum intensity. The normalized number is called the *response intensity* or simply the *response* of the neuron. Normalization is only for convenience. Non-normalized signal strengths, with the highest and lowest values labeled Max & Min rather than 1 and 0, would do as well.

The responses 1 and 0 are collectively referred to as binary signals and separately as high and low signals. For a signal consisting of spikes, a high signal consists of a burst of spikes at a high frequency. If 1 and 0 stand for the truth values TRUE and FALSE, neurons can process information contained in neural signals by functioning as logic operators.

For binary signals, the response of a neuron with one excitatory and one inhibitory input is assumed to be as shown in Table 1. Of the 16 possible binary functions of two variables, this table represents the only one that is consistent with the customary meanings of "excitation" and "inhibition." The table essentially says that a low excitatory input produces a low output signal (rows 1 and 2), a high excitatory input produces a high output (row 3), and a high inhibitory input suppresses a high excitatory input (row 4).

Some of the components in the figures require continuous, high input. This input is represented by the logic value "TRUE." For an electronic logic circuit, the high input is normally

**Table 1. Neuron response to binary inputs.** The table is also a logic truth table, with the last column representing the truth values of the statement X AND NOT Y.

| Excitatory X | Inhibitory Y | Response |
|:---:|:---:|:---:|
| 0 | 0 | 0 |
| 0 | 1 | 0 |
| 1 | 0 | 1 |
| 1 | 1 | 0 |

provided by the power supply. If the components represent neurons, the high input can be achieved by neurons in at least four ways. 1) A continuously high signal could be provided by a neuron that has excitatory inputs from many neurons that fire independently [27]. 2) Neurons that are active spontaneously and continuously without excitatory input are known to exist [28, 29]. A network neuron that requires a high excitatory input could receive it from a spontaneously active neuron, or 3) the neuron itself could be spontaneously active. 4) It will be seen that the high input could be provided by one of a flip-flop's outputs that is continuously high.

**4.2.2. Additive noise in binary neuron signals.** This section covers a potential problem for neural processing of digital (Boolean) information: Additive noise in binary inputs may affect the intended binary outputs of Table 1. The section includes three main points: Evidence indicates that some neurons have at least some rudimentary noise-reducing capabilities. For the NFF properties obtained here, noise can be sufficiently reduced by neurons that have two simple properties that generalize the noise-reducing properties of sigmoid functions. These properties do not indicate sophisticated capabilities.

*4.2.2.1. Noise reduction.* Two lines of evidence indicate that neurons have at least a minimal capability of reducing moderate levels of additive noise in binary inputs.

The persistent high and low firing frequency associated with short-term memory [14–16] and discussed above is itself evidence of a noise-reducing property. Without some noise-reducing capability, it would be difficult if not impossible for a network to maintain a variable output that can be either high or low. The cumulative effect of additive noise would quickly attenuate the output strength to a random walk through intermediate levels. This is the reason that simple noise-reducing nonlinearities are intentionally built into the materials in electronic components for digital signal processing, as demonstrated below by a single transistor's response.

Second, many neurons have sigmoid responses to single inputs, including inhibitory inputs [30–32]. In fact, "...the vast majority of neurons show sigmoid nonlinearities" [33]. A sigmoid response reduces moderate levels of additive noise in a binary input signal by producing an output that decreases a low input and increases a high input. It will be demonstrated by simulation that a neuron response that is sigmoid in both excitatory and inhibitory inputs is sufficient for the noise-reducing requirements of the NFFs presented here. But such a response is not necessary; a simpler, more general property is sufficient.

Reduction of noise in both excitatory and inhibitory inputs can be achieved by a response function of two variables that generalizes a sigmoid function's features. The noise reduction need only be slight for the proposed NFFs because they have feedback loops that continuously reduce the effect of noise.

Let F(X, Y) represent a neuron's response to an excitatory input X and an inhibitory input Y. The function must be bounded by 0 and 1, the minimum and maximum possible neuron responses, and must satisfy the values in Table 1 for binary inputs. For other points (X, Y) in the unit square, suppose F satisfies:

1. $F(X, Y) > X - Y$ for inputs (X, Y) near (1, 0) and

2. $F(X, Y) < X - Y$ or $F(X, Y) = 0$ for inputs (X, Y) near the other three vertices of the unit square.

The neuron responses of Table 1 are max{0, X-Y} (the greater of 0 and X-Y). For binary inputs with moderate levels of additive noise that makes them non-binary, conditions 1 and 2 make the output either closer to, or equal to, the intended output of Table 1 than max{0, X-Y}. Neurons that make up the networks proposed here are assumed to have these minimal noise-reducing properties.

Conditions 1 and 2 are sufficient to suppress moderate levels of additive noise in binary inputs and produce the NFF results found here. The level of noise that can be tolerated by the NFFs depends on the regions in the unit square where conditions 1 and 2 hold. If a binary input (X, Y) has additive noise that is large enough to change the region in which it lies, an error can occur.

*4.2.2.2. Example of a neuron response that satisfies conditions 1 and 2.* For any sigmoid function f from f(0) = 0 to f(1) = 1, the following function has the noise-reducing properties 1 and 2 and also satisfies Table 1:

$$F(X, Y) = f(X) - f(Y), \text{ bounded below by } 0.$$

This function is plausible as an approximation of a neuron response because it is sigmoid in each variable and some neurons are known to have sigmoid responses to single inputs, as mentioned above. The same sigmoid function applied to X and Y is not necessary to satisfy conditions 1 and 2. The function F could be the difference of two different sigmoid functions.

The function F is illustrated in Fig 2 for a specific sigmoid function f. The sine function of Fig 2A was chosen for f rather than any of the more common examples of sigmoid functions to demonstrate by simulation that a highly nonlinear function is not necessary for robust maintenance of binary signals. On half of the unit square, where Y ≥ X, Fig 2B shows that F has the value 0. This reflects the property that a large inhibitory input generally suppresses a smaller excitatory input.

*4.2.2.3. A primitive noise-reducing AND-NOT gate.* A response that satisfies conditions 1 and 2 in section 4.2.2.1 does not indicate capabilities of sophisticated logic or mathematics. An AND-NOT response with properties 1 and 2 can be produced by mechanisms that are quite simple. Fig 3 shows that a single transistor and three resistors can be configured to accomplish this. The network output was simulated in CircuitLab, and the graph was created in MS Excel and MS Paint. The inputs X and Y vary from 0V to 5V in steps of 0.05V. A 5V signal commonly stands for logic value 1, and ground stands for logic value 0.

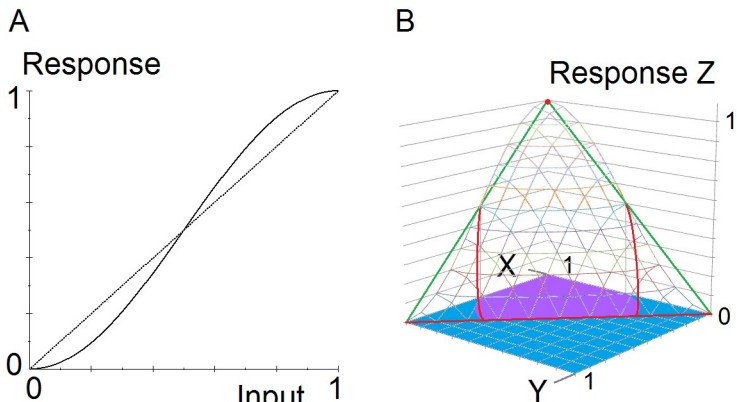

**Fig 2. Noise-reducing AND-NOT function.** The graphs show an example of a neuron response to analog inputs that reduces moderate levels of additive noise in binary inputs. **A.** A sigmoid function f(x) = (1/2)sin(π(x−1/2)) + 1/2. **B.** Graph of a function that has the noise-reducing properties 1 and 2. The function is F(X, Y) = f(X)—f(Y), bounded by 0. Wireframe: Graph of the response function Z = F(X, Y). Green and red: A triangle in the plane Z = X—Y. Red: Approximate intersection of the plane and the graph of F. Purple: Approximate region in the unit square where F(X, Y) > X—Y (condition 1). Blue: Approximate region in the unit square where F(X, Y) < X—Y or F(X, Y) = 0 (condition 2).

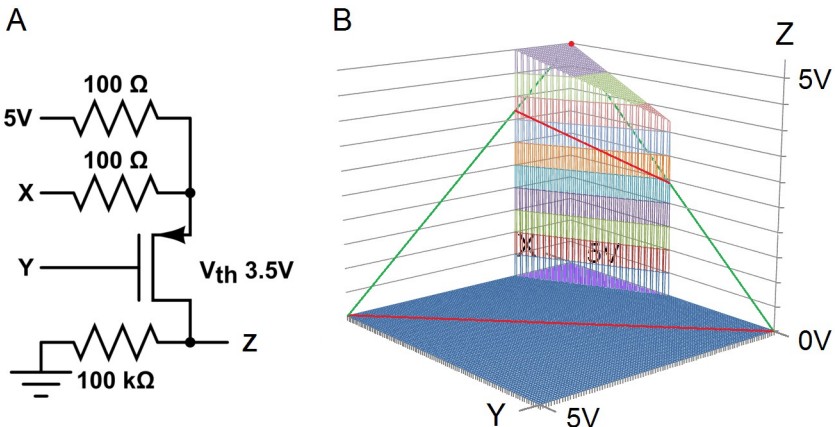

**Fig 3. Single transistor AND-NOT gate that reduces noise.** This minimal logic circuit satisfies the noise-reducing conditions 1 and 2. **A.** A logic circuit consisting of one transistor and three resistors. **B.** Engineering software simulation. Wireframe: Graph of the transistor response function Z = F(X, Y). Green and red: A triangle in the plane Z = X—Y. Red: Intersection of the plane and the graph of F. Purple: Region in the unit square where F(X, Y) > X—Y (condition 1). Blue: Region in the unit square where F(X, Y) < X—Y or F(X, Y) = 0 (condition 2).

## 4.3. Neural logic gates and flip-flops

Fig 4 shows three logic primitives and three flip-flops.

**4.3.1. Neural logic gates.** As discussed above, Fig 4A, consisting of an AND symbol and a NOT symbol, represents the logic function X AND NOT Y. The figure can also represent a neuron with one excitatory input and one inhibitory input, whose response to binary inputs is X AND NOT Y by Table 1. The logic outputs shown for Fig 4B and 4C also follow from the AND and NOT symbols.

The AND-NOT logic primitive has simplicity, efficiency, and power that have been under-appreciated. It is in the minority of logic primitives that are functionally complete. (As a technicality of logic, the AND-NOT operation is not functionally complete by itself because it requires access to the input TRUE to produce the NOT operation. Only the NAND and NOR operations are functionally complete by themselves. As a practical matter, NAND and NOR also require a high input for implementation.) Analogously to the single-neuron AND-NOT gate, the function can be implemented electronically with a single transistor and one resistor [5]. Any mechanism that can activate and inhibit like mechanisms and has access to a high activating input is a functionally complete AND-NOT gate. It may not be coincidence that the components of disparate natural signaling systems have these capabilities, e.g., immune system cells [34–37] and regulatory DNA [38, 39], in addition to transistors and neurons. As noted in the introduction, AND-NOT gates with analog signals can make up a powerful fuzzy logic decoder whose architecture is radically different from, and more efficient than, standard electronic decoder architectures [2, 4, 5]. Implemented with neural AND-NOT gates, these fuzzy decoders generate detailed neural correlates of the major phenomena of color vision and olfaction [1, 2].

**4.3.2. Neural flip-flops.** Fig 4 also shows three flip-flops. A flip-flop, or latch, is a common type of memory element used to store one bit of information in electronic computational systems. The more formal name is bistable multivibrator, meaning it has two stable states that can alternate repeatedly. A distinction is sometimes made between a "flip-flop" and a "latch," with the latter term reserved for asynchronous memory mechanisms that are not controlled by an oscillator. The more familiar "flip-flop" will be used here for all cases.

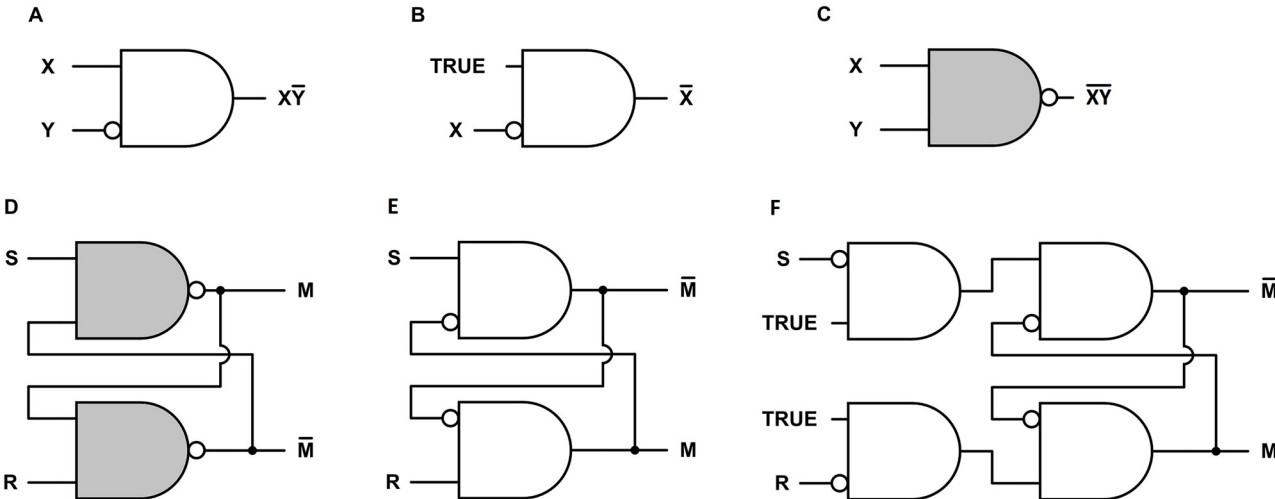

**Fig 4. Neural logic gates and flip-flops. A.** A symbol for an AND-NOT logic gate, with output X AND NOT Y. The symbol can also represent a neuron with one excitatory input X and one inhibitory input Y. **B.** An AND-NOT gate configured as a NOT gate, or inverter. **C.** A NAND gate (NOT AND). The output is NOT (X AND Y). There is no obvious way to implement this gate with a single neuron. **D.** A standard design for an electronic active low Set-Reset (SR) flip-flop composed of two NAND gates. **E.** An active low Set-Reset (SR) flip-flop composed of two AND-NOT gates. This design is derived from the design in **D** by moving each negation circle from one end of the connection to the other. This inverts the outputs. **F.** An active high SR flip-flop.

A flip-flop stores a discrete bit of information in an output with low and high values usually labeled 0 and 1. This output variable is labeled M in Fig 4. The value of M is the flip-flop *state* or *memory bit*. The information is stored by means of a brief input signal that activates or inactivates the memory bit. Input S *sets* the state to M = 1, and R *resets* it to M = 0. Continuous feedback maintains a stable state. A change in the state *inverts* the state.

Fig 4D shows a standard design for an electronic active low SR flip-flop. The S and R inputs are normally high. A brief low input S sets the memory bit M to 1, and a brief low input R resets it to 0. Fig 4E can be derived from Fig 4D simply by moving each negation circle from one end of the connection to the other. Importantly, this changes the logic of each component from NAND to AND-NOT, which can be implemented with a single neuron. The change only has one small effect on the network logic: If a circle is moved past an output, the output is inverted, as shown in Fig 4D and 4E.

Adding inverters to the inputs of Fig 4E produces the active high SR flip-flop of Fig 4F. The S and R inputs are normally low. A brief high input S sets the memory bit M to 1, and a brief high input R resets it to 0.

**4.3.3. Neural flip-flop simulation.** The simulation in Fig 5 demonstrates the robust operation of the NFF in Fig 4F in the presence of additive noise, using the neuron response function of Fig 2B in section 4.2.2.2. The simulation was done in MS Excel. The slow rise and fall of Set and Reset, over several delay times, is exaggerated to make the robust operation of the network clear.

Low level additive noise and baseline activity in the inputs are simulated by a computer-generated random number uniformly distributed between 0.01 and 0.1. The noise is offset by 0.01 so it does not obscure the high and low outputs in the graphs. The high Enabling input TRUE is simulated by 1 minus noise.

Each of the medium bursts in Set and Reset is simulated by the sum of two sine functions and the computer-generated noise. These signals could represent either noise bursts that are

not high enough to cause an error, or high input signals intended to invert the memory state but sufficiently reduced by noise to cause an error.

The two higher Set and Reset signals that invert the memory state are simulated by a sine function plus noise. These signals could represent either high input signals intended to invert the memory state, substantially reduced by noise but not enough to cause an error, or noise bursts with enough amplitude to cause an error.

The function F(X, Y) in Fig 2 was used for the simulated response of each NFF neuron as follows. The number $t_i$ represents the time after i neuron delay times, i = 0, 1, 2,. . .. At time $t_0$, the outputs are initialized at $M_0 = 0$ and $\bar{M}_0 = 1$. (If both are initialized at 0, they will oscillate until either Set or Reset is high.) At time $t_i$ for $i > 0$, the output $Z_i$ of each neuron that has excitatory and inhibitory inputs $X_{i-1}$ and $Y_{i-1}$ at time $t_{i-1}$ is:

$$Z_i = F(X_{i-1}, Y_{i-1})$$
$$= \max\{0, \ [(1/2)\sin(\pi(X_{i-1} - 1/2)) + 1/2] - [(1/2)\sin(\pi(Y_{i-1} - 1/2)) + 1/2]\}.$$

**4.3.4. Neural memory bank.** If information stored in short-term memory is no longer needed, active neurons consume energy without serving any useful purpose. An energy-saving function can be achieved with NFFs. Fig 6 shows a memory bank of three NFFs of Fig 4F, with a fourth serving as a switch to turn the memory bank on and off. The memory elements are enabled by excitatory input from the switch. A large memory bank could be organized as a tree, with switches at the branch potionints and memory elements at the leaves, so that at any time only the necessary memory elements are enabled.

# 5. Results

## 5.1. Plausibility of NFFs as short-term memory mechanisms

In all of the characteristics that are necessary for a mechanism to function as memory, as outlined in section 2.3, NFFs are plausible memory devices. Flip-flops are well understood and work well as memory devices in electronic computational systems. They are capable of storing different kinds of information. It is unlikely that any short-term memory mechanism in the brain could be simpler than the NFFs in Fig 4. The simulation shown in Fig 5 illustrates that NFFs can be robust in storing information. The decoders proposed in [2] can retrieve information held in NFFs. NFFs are efficient in material requirements (two or four neurons),

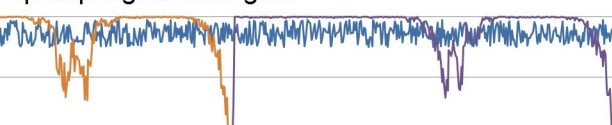

**Fig 5. Simulation of an NFF operation with noise in the inputs.** This simulation of the NFF in Fig 4F shows the NFF's operation is robust in the presence of moderate levels of additive noise in binary inputs. The effect of baseline noise on the memory bit is negligible, and temporary bursts of larger noise have no lasting effect.

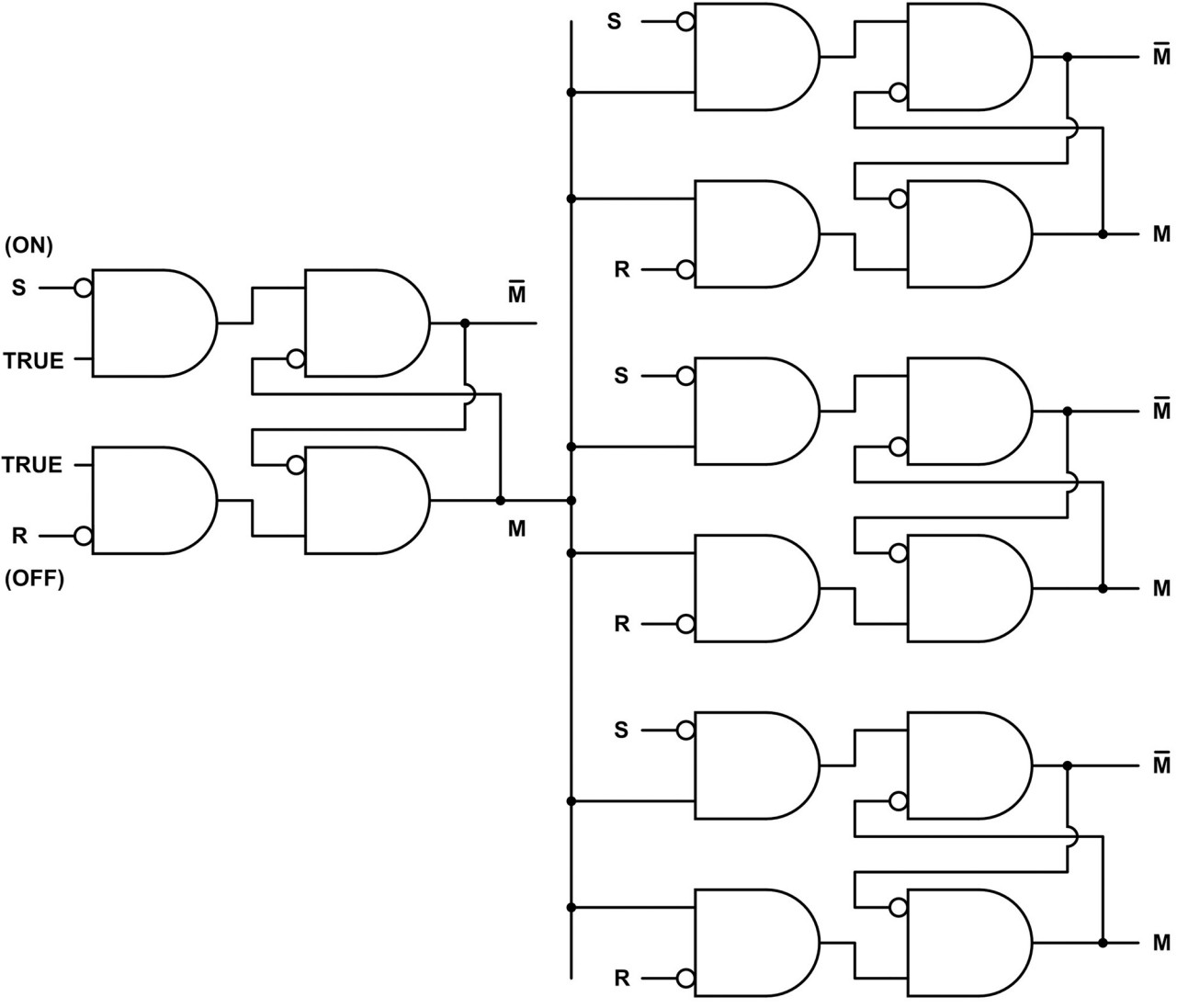

**Fig 6. Neural memory bank.** Three NFFs (Fig 4F) are enabled by a fourth NFF serving as an on-off switch.

operating requirements (no physical changes besides the level of neuron activity), and component capability requirements (excitation and inhibition). Because NFFs function dynamically, information can be stored quickly. The time required to set or reset an NFF is the time a signal takes to pass through two or three neurons, roughly 10–15 ms. The speed makes NFFs plausible models for short-term memory. NFFs consume energy continuously while they are holding information. This is consistent with the brain's high energy consumption, and it may be one of the selective pressures that resulted in static mechanisms for long-term memory.

## 5.2. Known memory phenomena generated by NFFs

NFF memory banks (Fig 6) can generate the seven characteristics of neuron firing that were listed in the section on unexplained memory phenomena. For all of the characteristics, one of the two outputs of an NFF in a memory bank is identical to the sampled neuron's response.

Since each NFF can store one bit of information, the number of NFFs that are required would depend on the amount of information to be recorded. To record the information conveyed by the stimulus, the visual, auditory, and sensorimotor cortexes would need to have neural structures to send the Set and Reset signals to the corresponding memory banks.

1. *Before the stimulus was presented, the sampled neuron discharged at a low, baseline level.* This describes one of the two NFF output neurons before the NFF state is inverted to record information. For convenience, label the output M before the NFF is set.

2. *When the stimulus was presented, or shortly after, the neuron began to fire at a high frequency.* This is the output M after the NFF is set by the input S.

3. *The high frequency firing continued after the stimulus was removed.* This is the stored memory bit M after the brief NFF input S returns to its normal state.

4. *The response was still high when the memory was demonstrated to be correct.* This is the high value of M holding information in memory as it is recalled.

5. *The response returned to the background level shortly after the test.* The memory bank (Fig 6) is turned off when the stored information is no longer needed, disabling all of the outputs.

6. *In the trials where the subject failed the memory test, the high level firing had stopped or* 7) *had never begun.* In instances where the high level firing had stopped, the memory bank was turned off before the memory was tested, or a distraction caused the NFF to be overwritten with new information, or noise or other errors inverted the NFF. In instances where the high level firing had never begun, the NFF was not set to record the information or the NFF recorded it incorrectly (for one of many possible reasons, e.g., the subject was not paying attention or was not motivated to remember). For each of these possibilities, the NFF would correctly predict both the failed memory test and the corresponding observed neuron behavior.

## 5.3. Limited capacity and duration of short-term memory

Compared to long-term memory, short-term memory can store only a small amount of information and only for a short time. The NFF model may provide at least a partial explanation for these limitations.

Short-term memory is short not only in duration but also in formation (a few milliseconds). Fast memory formation is an obvious biological advantage, even necessary for many mundane functions. That advantage could well have been the selective pressure that led to short-term memory. Speed means the mechanism must be dynamic in the sense that the only changes are the strengths of the neurons' signals.

Long-term memory is static. Memory models typically involve structural changes, such as neurogenesis, synaptogenesis, or pruning, or changes in synaptic strength or the strength of action potentials. Such changes may provide the large capacity and robust durability of long-term memory, but the changes are too slow for the fast formation of short-term memory.

Dynamic operation has the advantage of speed, but there are tradeoffs. First, NFFs' dynamic operation is expensive in energy use because at least one neuron in each NFF is highly active continuously while information is held in memory. This NFF activity is consistent with empirical evidence in short-term memory [14–16]. Second, the dynamic operation makes NFFs' stored information volatile. The state of an NFF can be inverted by temporary high additive noise in a set or reset input, a temporary loss of continuous energy input, or a temporary loss of a continuous high signal input. Neurons are notoriously unreliable in these aspects, which

would have a detrimental effect on the duration of memory implemented with NFFs. The cessation of the memory neuron activity before an error is made in a memory test is consistent with empirical evidence in short-term memory [14–16].

The memory bank in Fig 6 shows that many NFFs can be deactivated when information stored in short-term memory is no longer needed. This energy-saving function shortens the duration of short-term memory. The cessation of the memory neuron activity when information is no longer held in memory after a memory test is consistent with empirical evidence in short-term memory [14–16].

The capacity of short-term memory could be affected by its limited duration. The number of neurons devoted to short-term memory may be limited by the brain's ability to use information before the memory decays or becomes useless. If the prefrontal cortex cannot process large amounts of information conveyed by the senses at a sufficiently fast rate, there would be no reason to store that much information.

The capacity and duration of short-term memory could be affected by the difficulty in forming links. Long-term memory evidently relies on links for memory formation and retrieval. A name can be recalled from information associated with it. An extraordinary example of linkage is the number of songs that can be learned and recalled for decades, with seemingly effortless memory formation and recollection. A song's lyrics and melody are each connected linearly, and the two are connected to each other in parallel.

Short-term memory may not involve information that has such connections. Short-term memory often involves pseudorandom information (e.g., a phone number) that has no apparent possibility for a link. Arduous mnemonic techniques for linking unfamiliar items to familiar objects and places, such as "memory palaces," can increase the capacity of short-term memory somewhat, but such techniques fall far short of the capacity of long-term memory and have little or no effect on duration.

Other, unknown differences between long- and short-term memory may affect capacity or duration, such as how information is encoded in memory and how it is decoded.

## 5.4. Testable predictions

**5.4.1. Unknown memory phenomena generated by NFFs.** An NFF's outputs M and $\bar{\text{M}}$ together predict eight unknown phenomena that could further test whether short-term memory is produced by NFFs. These predictions can be tested by the same methods that were used in discovering the first seven phenomena since either M or $\bar{\text{M}}$ is predicted to be the output that produces those phenomena, and the other is predicted to be nearby.

1. *Along with the persistently active neuron associated with short-term memory* [14, 15], *another neuron has complementary output; i.e., when one is high the other is low*. This is predicted by M and $\bar{\text{M}}$ in the NFFs in Fig 4 and demonstrated in the simulation of Fig 5.

2. *The two neurons have reciprocal inputs*. This is shown in the NFFs in Fig 4.

3. *The two neurons are in close proximity*. This is because the neurons have reciprocal inputs and are part of a small network.

4. *The reciprocal inputs are inhibitory*. This is shown in the NFFs in Fig 4.

5. *The two neurons have some noise-reducing capability, such as responses that satisfy the inequalities 1 and 2*. Some noise-reducing capability is necessary to maintain robust binary outputs in the presence of additive noise.

6. *When the neurons change states, the high state changes first.* This is because the change in the neuron with the high output causes the change in the neuron with the low output. This can be seen in the NFFs in Fig 4, and it is demonstrated in the simulation of Fig 5. The change order is difficult to see in Fig 5 because of the time scale and the slow rise time of the Set and Reset inputs, but the simulation does have one neuron delay time between the completions of the two outputs' state changes.

7. *The other neuron's output then changes from low to high within a few milliseconds.* This happens quickly because reciprocal input from the first change causes the second within approximately one neuron delay time, regardless of how long information is held in memory.

8. *After the memory test, the outputs of both neurons are low.* The memory bank (Fig 6) is turned off when the stored information is no longer needed, disabling all of the outputs.

**5.4.2. Predicted behavior of constructed neural networks.** Any of the networks in Fig 4 or the memory bank of Fig 6 could be constructed with neurons and tested for predicted behavior. If the single neuron in Fig 4A produces the outputs of Table 1, then the predicted operations of all of the networks should follow. The NFFs are predicted to have stable outputs that are inverted by a brief input from S or R. (Recall the NFF of 4E is active low.) The outputs should also exhibit the properties predicted for NFFs in the preceding section.

## Acknowledgments

Simulations were done with MS Excel and CircuitLab. Network diagrams were created with CircuitLab and MS Paint. Graphs were created with Converge 10.0, MS Excel, and MS Paint. The author would like to thank Arturo Tozzi, David Garmire, Robert Barfield, Paul Higashi, Anna Yoder Higashi, Sheila Yoder, and especially Ernest Greene and David Burress for their support and many helpful comments and suggestions.

## Author Contributions

**Conceptualization:** Lane Yoder.

**Data curation:** Lane Yoder.

**Formal analysis:** Lane Yoder.

**Funding acquisition:** Lane Yoder.

**Investigation:** Lane Yoder.

**Methodology:** Lane Yoder.

**Project administration:** Lane Yoder.

**Resources:** Lane Yoder.

**Software:** Lane Yoder.

**Supervision:** Lane Yoder.

**Validation:** Lane Yoder.

**Visualization:** Lane Yoder.

**Writing – original draft:** Lane Yoder.

**Writing – review & editing:** Lane Yoder.

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
