## [Decision Letter · Decision Letter 0]

28 Feb 2023

PONE-D-22-30620Neural Flip-Flops I: Short-Term MemoryPLOS ONE

Dear Dr. Lane Yoder

Thank you for submitting your manuscript to PLOS ONE. After careful consideration, we feel that it has merit but does not fully meet PLOS ONE’s publication criteria as it currently stands. Therefore, we invite you to submit a revised version of the manuscript that addresses the points raised during the review process.

We look forward to receiving your revised manuscript.

Kind regards,

Wazir Muhammad

Academic Editor

PLOS ONE

Journal Requirements:

Additional Editor Comments:

Dear author at present we cannot accept your paper, you should revise your paper draft according to reviewer comments.

Reviewers' comments:

Reviewer's Responses to Questions

**Comments to the Author**

1. Is the manuscript technically sound, and do the data support the conclusions?

Reviewer #1: Yes

2. Has the statistical analysis been performed appropriately and rigorously? 

Reviewer #1: Yes

3. Have the authors made all data underlying the findings in their manuscript fully available?

Reviewer #1: Yes

4. Is the manuscript presented in an intelligible fashion and written in standard English?

Reviewer #1: No

5. Review Comments to the Author

Reviewer #1: The manuscript entitled “Neural Flip-Flops I: Short-Term Memory” has been investigated in detail. The topic addressed in the manuscript is potentially interesting and the manuscript contains some practical meanings, however, there are some issues which should be addressed by the authors:

1) In the first place, I would encourage the authors to extend the abstract more with the key results. As it is, the abstract is a little thin and does not quite convey the interesting results that follow in the main paper. The "Abstract" section can be made much more impressive by highlighting your contributions. The contribution of the study should be explained simply and clearly.

2) The readability and presentation of the study should be further improved. The paper suffers from language problems.

3) The importance of the design carried out in this manuscript can be explained better than other important studies published in this field. I recommend the authors to review other recently developed works.

4) “Discussion” section should be added in a more highlighting, argumentative way. The author should analysis the reason why the tested results is achieved.

5) The authors should clearly emphasize the contribution of the study. Please note that the up-to-date of references will contribute to the up-to-date of your manuscript. The study named- Crude oil time series prediction model based on LSTM network with chaotic Henry gas solubility optimization- can be used to explain the method in the study or to indicate the contribution in the “Introduction” section.

6) How to set the parameters of proposed method for better performance?

7) It will be helpful to the readers if some discussions about insight of the main results are added as Remarks.

This study may be proposed for publication if it is addressed in the specified problems.

6. PLOS authors have the option to publish the peer review history of their article (what does this mean?). If published, this will include your full peer review and any attached files.

Reviewer #1: No

---

## [Author Response · Author response to Decision Letter 0]

25 Aug 2023

Please see the Response to the Reviewers.

---

## [Decision Letter · Decision Letter 1]

15 Dec 2023

PONE-D-22-30620R1Neural Flip-Flops I: Short-Term MemoryPLOS ONE

Dear Dr. Yoder,

Thank you for submitting your manuscript to PLOS ONE. After careful consideration, we feel that it has merit but does not fully meet PLOS ONE’s publication criteria as it currently stands. Therefore, we invite you to submit a revised version of the manuscript that addresses the points raised during the review process.

We look forward to receiving your revised manuscript.

Kind regards,

Sunder Ali Khowaja, Ph.D.

Academic Editor

PLOS ONE

**Additional Editor Comments:**

Dear Authors,

The reviewers have completed their review for this paper. One of the reviewers have shown concerns regarding the novelty and methodology of the proposed work. Authors are requested to prepare a thorough response for the next round of review.

Reviewers' comments:

Reviewer's Responses to Questions

**Comments to the Author**

1. If the authors have adequately addressed your comments raised in a previous round of review and you feel that this manuscript is now acceptable for publication, you may indicate that here to bypass the “Comments to the Author” section, enter your conflict of interest statement in the “Confidential to Editor” section, and submit your "Accept" recommendation.

Reviewer #1: All comments have been addressed

Reviewer #2: (No Response)

Reviewer #3: (No Response)

2. Is the manuscript technically sound, and do the data support the conclusions?

Reviewer #1: Yes

Reviewer #2: Partly

Reviewer #3: Yes

3. Has the statistical analysis been performed appropriately and rigorously? 

Reviewer #1: Yes

Reviewer #2: I Don't Know

Reviewer #3: Yes

4. Have the authors made all data underlying the findings in their manuscript fully available?

Reviewer #1: Yes

Reviewer #2: No

Reviewer #3: Yes

5. Is the manuscript presented in an intelligible fashion and written in standard English?

Reviewer #1: Yes

Reviewer #2: Yes

Reviewer #3: Yes

6. Review Comments to the Author

Reviewer #1: The authors have discussed the issues I mentioned in detail. It is appropriate for the study to be accepted for publication.

Reviewer #2: The author has produced a simulation model based on the use of neuronal flipflops to describe several features of short-term memory. The concept is based on simple logic of 0/1 switches and the relative simplicity with which these symptoms can store information and provide a context for maintenance of information in short-term memory. I am reviewing this paper after it has already been reviewed by one reviewer who seems supportive of the overall paper itself but requested clarification.

As a neuroscientist rather than a computer scientist or mathematician, I generally find papers like this with no value in understanding brain function in any meaningful way. The system described here is completely artificial and really has no relevance to the way normal neurons interact. There is no synaptic pruning, no molecular changes, and as is well known, input to neurons is highly complex with thousands of inputs being provided. There are change in firing rates, firing patterns, oscillatory inputs that cause phase locking and spike-field coherence, etc that are all need to understand complex processes such as memory formation storage, recall and so on. Models such as this are potentially useful in trying to provide framework for understanding brain function but, in reality they are tools for the design of artificial neuro networks.

As I am not an expert in logic systems, it is difficult for me to comment on the rigor here, although based on my experience in computational neuroscience, I would say that the neuronal model here is exceptionally simplistic. I understand that this is one of the goals of the author- to indicate that a very simple system that can do powerful things- but this is by no means a new observation. Basic two and three layer neural networks that integrate inputs from input layers to produce a flip-fop like response have been able to perform such tasks for decades and continue to evolve in their complexity despite using fairly simple logic systems for decision making. From a pure neuroscience point of view, I would argue that this paper has relatively low value, although from a computational and engineering perspective, this may not be the case and therefore, I would defer on this aspect.

Regarding the paper’s substance, I would make a few comments. First, it is exceptionally long. It reads much more like a textbook than a research paper and a typical reader would get lost in the extensive descriptions of logic systems. This paper should be much more focused in the actual results, which in my opinion are very difficult to discern from discussion about the results, which are intermixed. Second, I did not see a real example of a discreet case in which a memory is learned, recalled, etc. The author described case examples, but these seems to be presented more as a theoretical response. The reader has no idea what a specific memory test would look like.

.

Reviewer #3: The paper considers "neural flip-flops", bistable circuits of neurons, as possible devices for short-term memory in brain. Such flip-flops are thoroughly studied and widely used in electronics, so the only achievement of the paper is the assumption that similar systems can be relevant in neuroscience as well. There is no surprise that one can come up with a simple circuit which demonstrates bistable behaviour, but I highly doubt that circuits similar to NFFs are used in working memory of real brains. The most obvious reason is that at least one neuron of the NFF is always ON which is very expensive in terms of power. The other reason is that the brain functions barely rely on such small circuits due to low reliability of single neurons. After all, such a simple model does not suggest any explanations for the key characteristics of the working memory: capacity and duration, which can be unlimited in a population of NFFs. For the above reasons I do not see much scientific value in the present study and can not support its publication.

7. PLOS authors have the option to publish the peer review history of their article (what does this mean?). If published, this will include your full peer review and any attached files.

Reviewer #1: No

Reviewer #2: No

Reviewer #3: No

---

## [Author Response · Author response to Decision Letter 1]

17 Dec 2023

The response to the reviewers is in the uploaded file.

---

## [Decision Letter · Decision Letter 2]

2 Feb 2024

PONE-D-22-30620R2Neural Flip-Flops I: Short-Term MemoryPLOS ONE

Dear Dr. Yoder,

Thank you for submitting your manuscript to PLOS ONE. After careful consideration, we feel that it has merit but does not fully meet PLOS ONE’s publication criteria as it currently stands. Therefore, we invite you to submit a revised version of the manuscript that addresses the points raised during the review process.

The reports from the reviewers have been submitted. It is observed that one of the reviewers still have concerns regarding the quality of the work as well as some of their comments which authors did not respond to. Authors are requested to incorporate the reviewer comments and prepare a point-by-point response for the reviewers for a quick assessment.

We look forward to receiving your revised manuscript.

Kind regards,

Sunder Ali Khowaja, Ph.D.

Academic Editor

PLOS ONE

Additional Editor Comments:

The reports from the reviewers have been submitted. It is observed that one of the reviewers still have concerns regarding the quality of the work as well as some of their comments which authors did not respond to. Authors are requested to incorporate the reviewer comments and prepare a point-by-point response for the reviewers for a quick assessment.

Reviewers' comments:

Reviewer's Responses to Questions

**Comments to the Author**

1. If the authors have adequately addressed your comments raised in a previous round of review and you feel that this manuscript is now acceptable for publication, you may indicate that here to bypass the “Comments to the Author” section, enter your conflict of interest statement in the “Confidential to Editor” section, and submit your "Accept" recommendation.

Reviewer #1: All comments have been addressed

Reviewer #3: All comments have been addressed

2. Is the manuscript technically sound, and do the data support the conclusions?

Reviewer #1: Yes

Reviewer #3: (No Response)

3. Has the statistical analysis been performed appropriately and rigorously? 

Reviewer #1: Yes

Reviewer #3: I Don't Know

4. Have the authors made all data underlying the findings in their manuscript fully available?

Reviewer #1: Yes

Reviewer #3: Yes

5. Is the manuscript presented in an intelligible fashion and written in standard English?

Reviewer #1: Yes

Reviewer #3: Yes

6. Review Comments to the Author

Reviewer #1: My comments have been addressed. It is acceptable in the present form.

Reviewer #3: The author has addressed most of my concerns in a convincing way. I still find that the results of the paper are of small value, but this is my subjective assessment. The authors claims that he is the first to assume that local bistable circuits can serve as the memory basis in the brain, and this clain I can not check or refute.

I would still like the Author to reply on my last comment regarding the capacity and duratino of the working memory. It is known that it can store a limited number of items (about 7) for a limited amount of time (tens of seconds). How can these properties be explained by the suggested model?

7. PLOS authors have the option to publish the peer review history of their article (what does this mean?). If published, this will include your full peer review and any attached files.

Reviewer #1: No

Reviewer #3: No

---

## [Author Response · Author response to Decision Letter 2]

11 Feb 2024

The following subsection was added in the revision on page 26.

5.3. Limited capacity and duration of short-term memory

---

## [Decision Letter · Decision Letter 3]

29 Feb 2024

Neural Flip-Flops I: Short-Term Memory

PONE-D-22-30620R3

Dear Dr. Yoder,

We’re pleased to inform you that your manuscript has been judged scientifically suitable for publication and will be formally accepted for publication once it meets all outstanding technical requirements.

Kind regards,

Sunder Ali Khowaja, Ph.D.

Academic Editor

PLOS ONE

Additional Editor Comments (optional):

Reviewers' comments:

Reviewer's Responses to Questions

**Comments to the Author**

1. If the authors have adequately addressed your comments raised in a previous round of review and you feel that this manuscript is now acceptable for publication, you may indicate that here to bypass the “Comments to the Author” section, enter your conflict of interest statement in the “Confidential to Editor” section, and submit your "Accept" recommendation.

Reviewer #1: All comments have been addressed

Reviewer #3: All comments have been addressed

2. Is the manuscript technically sound, and do the data support the conclusions?

Reviewer #1: Yes

Reviewer #3: Yes

3. Has the statistical analysis been performed appropriately and rigorously? 

Reviewer #1: Yes

Reviewer #3: Yes

4. Have the authors made all data underlying the findings in their manuscript fully available?

Reviewer #1: No

Reviewer #3: Yes

5. Is the manuscript presented in an intelligible fashion and written in standard English?

Reviewer #1: Yes

Reviewer #3: Yes

6. Review Comments to the Author

Reviewer #1: (No Response)

Reviewer #3: All my concerns are addressed, the author have added the necessary explanations, the paper can be published

7. PLOS authors have the option to publish the peer review history of their article (what does this mean?). If published, this will include your full peer review and any attached files.

Reviewer #1: No

Reviewer #3: No

---

## [Editor Report · Acceptance letter]

5 Mar 2024

PONE-D-22-30620R3 

PLOS ONE

Dear Dr. Yoder, 

I'm pleased to inform you that your manuscript has been deemed suitable for publication in PLOS ONE. Congratulations! Your manuscript is now being handed over to our production team.

Kind regards, 

on behalf of

Dr. Sunder Ali Khowaja 

Academic Editor

PLOS ONE